

# Regulation of S1P receptors and sphingosine kinases expression in acute pulmonary endothelial cell injury

Huiying Liu[1], Zili Zhang[1], Puyuan Li[1], Xin Yuan[1], Jing Zheng[1], Jinwen Liu[2], Changqing Bai[1] and Wenkai Niu[1]

[1] Department of Respiratory and Critical Care Diseases, 307th Hospital of PLA, Beijing, The People's Republic of China
[2] Beijing Oriental Yamei Gene Science & Technology Institute, Beijing, The People's Republic of China

## ABSTRACT

**Background**. Acute lung injury and acute respiratory distress syndrome (ALI/ARDS) is a severe clinical syndrome with mortality rate as high as 30–40%. There is no treatment yet to improve pulmonary endothelial barrier function in patients with severe pulmonary edema. Developing therapies to protect endothelial barrier integrity and stabilizing gas exchange is getting more and more attention. Sphingosine-1-phosphate (S1P) is able to enhance the resistance of endothelial cell barrier. S1P at physiological concentrations plays an important role in maintaining endothelial barrier function. Proliferation, regeneration and anti-inflammatory activity that mesenchymal stem cells (MSCs) exhibit make it possible to regulate the homeostatic control of S1P.

**Methods**. By building a pulmonary endothelial cell model of acute injury, we investigated the regulation of S1P receptors and sphingosine kinases expression by MSCs during the treatment of acute lung injury using RT-PCR, and investigated the HPAECs Micro-electronics impedance using Real Time Cellular Analysis.

**Results**. It was found that the down-regulation of TNF-$\alpha$ expression was more significant when MSC was used in combination with S1P. The combination effection mainly worked on S1PR2, S1PR3 and SphK2. The results show that when MSCs were used in combination with S1P, the selectivity of S1P receptors was increased and the homeostatic control of S1P concentration was improved through regulation of expression of S1P metabolic enzymes.

**Discussions**. The study found that, as a potential treatment, MSCs could work on multiple S1P related genes simultaneously. When it was used in combination with S1P, the expression regulation result of related genes was not simply the superposition of each other, but more significant outcome was obtained. This study establishes the experimental basis for further exploring the efficacy of improving endothelial barrier function in acute lung injury, using MSCs in combination with S1P and their possible synergistic mechanism.

Corresponding authors
Changqing Bai, mlp1604@sina.com
Wenkai Niu, niuwk88@sina.com

# INTRODUCTION

Acute lung injury and acute respiratory distress syndrome (ALI/ARDS) was first recognized as a clinical syndrome in the 1960s. It manifests as severe and acute respiratory failure with hypoxemia and lung infiltration and is often caused by pneumonia, sepsis and major trauma. The mortality rate of ALI/ARDS is as high as 30–40% (*Rubenfeld et al., 2005*). In the large number of drugs evaluated by clinical trials, none has proven to be effective or could be recommended as the standard treatment of ALI/ARDS (*Cepkova & Matthay, 2006*). Supportive therapy is currently the major treatment, including protective ventilation and conservative medication (*Brower et al., 2000*; *Wiedemann et al., 2006*). Mechanical ventilation is a necessary and life-saving method, but it may delay the inflammatory response and ultimately results in pulmonary endothelial barrier dysfunction. Endothelial barrier dysfunction can lead to increased permeability, extravasation of fluid rich in proteins and pulmonary edema, which are all common symptoms of ALI/ARDS (*Kumar et al., 2008*). There is currently no treatment to improve pulmonary endothelial barrier function in severe pulmonary edema patients (*Müller-Redetzky, Suttorp & Witzenrath, 2014*). Developing a therapy to protect endothelial barrier integrity and stabilize gas exchange is getting a lot of attention.

Sphingosine-1-phosphate (S1P) is a ubiquitous sphingomyelin and is an important regulator of vascular endothelial cell permeability and fluid balance *in vivo*. It is mainly present in plasma and tissues and is able to enhance the resistance of endothelial barrier. Endothelial barrier enhancement mediated by S1P is completed by activating the Gi and Racl signaling pathway through S1P receptor (*Sun et al., 2012*). Studies have demonstrated that S1P plays an important role in allergic reactions in the respiratory system. S1P promotes the adhesion of endothelial cells, which is the key in maintaining endothelial barrier and avoiding increased permeability leading to pulmonary edema (*Lee et al., 1999*). FTY720, structure analogue of S1P, was approved by FDA in the treatment of multiple sclerosis in 2010. FTY720 slightly differs from S1P in receptor binding activity, but they both have strong side effects at high doses (*Natarajan et al., 2013*). This indicates that in the treatment of lung disease, selectivity of S1P receptors and homeostatic control of S1P concentration is important.

Several studies have demonstrated that S1P, at physiological concentration, has an important role in maintaining endothelial barrier function. For S1P and its structure analogues, their limitations in therapeutic function, confirm the conception of sphingolipid homeostasis. The fate of the cells is determined by the homeostasis of S1P concentration (*Cuvillier et al., 1996*). As an endogenous bioactive molecule, S1P is an important regulator of vascular endothelial cell permeability and fluid balance. Its anabolic and key molecules on signaling pathways are still potential targets and focus of research in the treatment of ALI/ARDS.

Recent clinical studies have found that it is unlikely for any single drug to reverse the severe pathological injury caused by acute lung injury. Therefore, in the therapy of ALI/ARDS, there has been attention focused on cell therapy, especially mesenchymal stem cell therapy.

Mesenchymal stem cells (MSCs) are multi-functional cells with the ability of self-renewal. They were first discovered in bone marrow and can differentiate into bone, cartilage, fat, muscle and other tissues. At present, MSCs have demonstrated great efficacy in the treatment of cirrhosis (*Zheng et al., 2012*), systemic lupus erythematosus (*Wang et al., 2013*) and other diseases. Therapeutic effects have also been obtained in preclinical studies of Crohn's disease (*Valcz et al., 2011*), traumatic brain injury (*Harting et al., 2009*), sepsis (*Nemeth et al., 2008*), acute renal failure (*Bruno et al., 2009*) and other diseases. In recent years, several studies have reported, using a number of animal models of lung disease that allograft MSC therapy can reduce lung injury? (*Chang et al., 2009*; *Moodley et al., 2009*; *Ortiz et al., 2003*)

Most recent studies involving MSCs in the treatment of ALI/ARDS are focused on the evaluation of efficacy, with no further discussion of the immunomodulatory impact of MSCs on pulmonary tissue cells, especially the endothelial cell barrier. The regulation of S1P metabolism by MSCs is lacking in many systematic and comprehensive studies too. Homeostatic control of S1P *in vivo* is a prerequisite to improving endothelial barrier and treating acute lung injury. The anabolic process of S1P is regulated by many cytokines. The outcomes of many diseases when MSCs were used demonstrate that it can secrete various growth and inflammatory factors, playing a role in immune regulation. Besides, according to some studies, S1P is capable of promoting the differentiation of MSCs *in vitro* (*He et al., 2010*). This suggests the possible synergistic mechanism of MSCs and S1P in the treatment of ALI/ARDS. On the one hand, while S1P plays a role in enhancing endothelial barrier function, MSCs modulates the host's immune response to injury through its proliferation, regeneration and anti-inflammatory effects. On the other hand, with the differentiation accelerating effect of S1P on MSCs, the MSCs have better immune regulation. The effect of MSC on the expression of S1P receptors and metabolic enzymes, leads to better *in vivo* homeostasis of S1P which in turn enhances the endothelial barrier.

Therefore in this study, by building an acute injury model of pulmonary endothelial cells induced by LPS, the regulatory effect of MSCs on the expression of S1P receptors and sphingosine kinase, in the treatment of acute lung injury was investigated. The efficacy of improving endothelial barrier function in acute lung injury when MSCs were used in combination with S1P and their possible synergistic mechanism are discussed. Thus it provides theoretical and experimental basis for the treatment of acute lung injury.

## MATERIALS AND METHODS

### Standards and reagents
Sphingosine-1-Phosphate (S1P) purchased from Sigma-Aldrich Company was dissolved in methanol and kept at $-20\,°C$. Lipopolysaccharide (LPS) E.coli O55:B5 purchased from Sigma-Aldrich Company was dissolved in saline and kept at $-20\,°C$.

### Antibody
Anti-EDG-1 (S1PR1), Anti-EDG-5 (S1PR2) and Anti-EDG-3 (S1PR3) were purchased from Santa Cruz Biotechnology. Anti-$\beta$-actin was purchased from Sigma-Aldrich.

## Cell culture

Newborn umbilical cord was taken under sterile conditions (The Affiliated Hospital of Military Medical Science Scientific Research Ethics Committee Approval Report, No ky-2015-3-17) and remnant blood in umbilical vein and artery was rinsed in D-HANKS solution containing $10^5$U/L penicillin and 100 mg/L streptomycin. Collagenase IV solution was added to digest the cells, at 37 °C for 2 h. Single cell suspension of MSCs was obtained following filtration. After the cells were counted, the suspension was inoculated into 10 ml MSC medium (Thermo Fisher, Waltham, MA, USA) in a 100 mm petri dish with a cell density of 2.5 to $4 \times 10^4$/cm$^2$. After primary culture at 37 °C and 5% $CO_2$ in an incubator for 7 to 9 days, first passage was carried out according to overall growth condition and local density. Incubation was continued until the density exceeds 80%, repeated the above operations. Subculture generations at P3 to P5 were used for subsequent experiments. Human pulmonary artery endothelial cells (HPAECs) purchased from ScienCell Company were cultured in endothelial cell medium (ScienCell, Carlsbad, CA, USA) at 37 °C in a 5% $CO_2$ in incubator. P5 to P8 cells were used for subsequent experiments.

## Acute Injury of cells

HPAEC cells between P5 to P8 were inoculated into a 16-well E-Plate (ACEA Biosciences, San Diego, CA, USA) at a concentration of $2 \times 10^4$ per well in 100 µl, then cultured in an incubator at 37 °C and 5% $CO_2$ for 12 h following which LPS was added at varying concentrations of 0, 0.5, 1, 2 and 5 µM. The 16-well E-Plates were placed within the Real Time Cellular Analysis (RTCA) System (ACEA Biosciences) and cultured in an incubator at 37 °C and 5% $CO_2$. Micro-electrical impedance of HPAEC cells were detected in real-time to investigate the effects of acute injury caused by LPS on HPAEC cells, at different concentrations.

HPAEC cells between P5 to P8 were inoculated into a 24-well cell culture plate (Corning, Inc., Corning, NY, USA) at a concentration of $1 \times 10^5$ per well in 600 µl, then cultured in incubator at 37 °C and 5% $CO_2$ for 6 to 12 h after LPS was added at a final concentration of 1 µM. Cells were collected at different time points and the expression of TNF-$\alpha$ was detected to investigate the acute injury effects of LPS on HPAEC cells.

## Cell co-culture

HPAEC cells between P5 to P8 were inoculated into a 16-well E-Plate (ACEA Biosciences) in 100 µl at a concentration of $2 \times 10^4$ per well and then LPS at a concentration of 1 µM was added. In the meantime, 60 µl of MSCs at different proportions (wherein the ratio of HPAEC and MSC were 1:1, 1:2, and 1:4, the control group is MSCs medium only) were inoculated into a 16-well E-Plate Insert (ACEA Biosciences), The insert was then placed into a receiver plate containing 100 µl of MSC medium. The 16-well E-Plates were placed within the RTCA System, together with the receiver plates, and were cultured in an incubator at 37 °C and 5% $CO_2$ for 12 h. Following that, medium in the E-Plates was discarded and replaced with 100 µl of fresh endothelial cell medium, and the insert was filled to 60 µl with MSC medium in the corresponding receiver plates. The inserts were then placed into the E-Plates containing HPAECs and co-cultured within the RTCA System

at 37 °C and 5% $CO_2$ for 8–24 h. Micro-electrical impedance of HPAEC cells were detected in real-time to investigate the effect of MSCs on HPAEC cell injury.

HPAEC cells between P5 to P8 in 600 μl medium were inoculated into a 24-well cell culture plate at a concentration of $1 \times 10^5$ per well following which LPS at a concentration of 1 μM was added. Meanwhile, 100 μl of MSCs with different proportions (wherein the ratio of HPAEC and MSC were 1:1, 1:2, and 1:4, the control group is MSCs medium only) were inoculated into the upper compartment of a 24 well Transwell plate (0.4 μm Polyester Membrane; Corning, Inc., Corning, NY, USA), and 600 μl of MSC medium was added into the lower compartment. The 24-well cell culture plates and Transwell plates were incubated at 37 °C and 5% $CO_2$ for 12 h. Then the medium in the cell culture plates was discarded and replaced with 600 μl of fresh endothelial cell medium, the upper compartment of the Transwell plates was filled to 100 μl with MSC medium in the corresponding lower compartments. After 8 h to 24 h of co-culture with the upper compartment of Transwell plates placed into cell culture plates, HPAEC cells were collected. The effect of MSCs on HPAEC cell injury was investigated by determining the expression change of TNF-$\alpha$.

## Detection of Cell Micro-electronics impedance
A 16-well E-Plate was placed on Xcelligence RTCA DP system (ACEA Biosciences) and incubated at 37 °C and 5% $CO_2$. Detection was determined according to the manufacturer's instructions.

## Real-time quantitative PCR
Total RNA was extracted from collected HPAEC cells using RNeasy® Mini Kit (Qiagen, Germantown, MD, USA) according to manufacturer's instructions. With 1 μg of RNA as template, reverse transcription reaction was performed using QuantScript RT Kit (TIANGEN, Beijing, China) according to manufacturer's instructions. Utilizing Bio-Rad iQ$^{TM}$5 Multicolor Real-Time PCR Detection System and with GAPDH as internal reference, fluorescence intensity was detected using SuperReal PreMix SYBR Green (TIANGEN, Beijing, China) according to the manufacturer's instructions. The primers used are shown in Table 1. Relative expression level of mRNA was calculated by 2-$\Delta\Delta$Ct method.

## Western blot analysis
The collected HPAEC cells were washed with PBS twice. Then 200 μl lysis buffer was added, which contains 20 mM Tris–HCl (pH 7.5), 150 mM NaCl, 1 mM EDTA, 1 mM EGTA, 1% Triton X-100, 2.5 mM pyrophosphate sodium phosphate, 1 μg/ml leupeptin, 1 μg/ml aprotinin and protease inhibitors (Roche, Indianapolis, IN, USA). Cell lysate was centrifuged at $10,000 \times g$, room temperature for 10 min and the supernatant was collected. After protein concentration was measured using BCA method, 20 to 50 μg of protein was separated by 12% SDS-PAGE, then transferred to a PVDF membrane and sealed with TBST buffer containing 5% BSA. Specific antibodies were added and incubated overnight. On the next day, after washing and secondary antibody incubation, chromogenic assay was performed using ECL chemiluminescence kit (GE Healthcare, Chicago, IL, USA).

**Table 1  Primer sequences.**

| Primer | Sequence |
| --- | --- |
| S1PR1-F | 5′-GCACCAACCCCATCATTTAC-3′ |
| S1PR1-R | 5′-TTGTCCCCTTCGTCTTTCTG-3′ |
| S1PR2-F | 5′-CAAGTTCCACTCGGCAATGT-3′ |
| S1PR2-R | 5′-CAGGAGGCTGAAGACAGAGG-3′ |
| S1PR3-F | 5′-TCAGGGAGGGCAGTATGTTC-3′ |
| S1PR2-R | 5′-GAGTAGAGGGGCAGGATGGT-3′ |
| SphK1-F | 5′-TCTGGGCACCTTCCTGCGTC-3′ |
| SphK1-R | 5′-CTCACTGCCCAGGTGCGAGTG-3′ |
| SphK2-F | 5′-TGCTGGAAGGTGGGCGTC-3′ |
| SphK2-R | 5′-AATAGACTCCGCCCTCAGCC-3′ |
| TNFα-F | 5′-TGATCCCTGACATCTGGAATCTG-3′ |
| TNFα-R | 5′-GCCAAGGTCCACTTGTGTC-3′ |
| GAPDH | 5′-GAAGGTGAAGGTCGGAGTC-3′ |
| GAPDH | 5′-GAAGATGGTGATGGGATTTC-3′ |

## Statistical analysis

All experimental data are represented as mean ± standard deviation (mean ± SD). One-way ANOVA (Dunnett test) was adopted in multiple group comparison. Data was analyzed with software GraphPad Prism 6.0. $p < 0.05$ indicates that the difference is statistically significant.

## RESULTS

### Acute injury can be caused 12 h after exposure of HPAECs to 1 μM LPS

LPS is widely used in the preparation of acute lung injury model. However, there is no standard procedure that has been developed. Literatures shows that acute injury induced by LPS will cause significant endothelial cell barrier disorder and increase in permeability. In the meantime, there is abnormal expression of a variety of growth and inflammatory factors. Particularly, the expression of TNFα is significantly increased (*Meduri et al., 2009*). Here HPAECs were stimulated using LPS at varying final concentrations of 0, 0.5, 1, 2 and 5 μM for 12 h, respectively. The micro-electrical impedance changes of endothelial cells were detected by Real Time Cellular Analysis (RTCA) to reflect the changes in endothelial barrier function (Fig. 1A). Results show that compared to other concentrations, micro-electrical impedance dropped significantly when 1 μM LPS was added to HPAECs (Fig. 1B). This indicates that 1 μM LPS is more effective in acute injury. Furthermore, HPAECs were stimulated by LPS at a concentration of 1 μM for different time to detect the change in TNF-α expression. Results demonstrated that the expression of TNFα was significantly increased at 12 h after stimulation (Fig. 1C). This suggests that HPAECs stimulated by 1 μM LPS for 12 h can produce an acute cell injury model for subsequent studies.

**A**

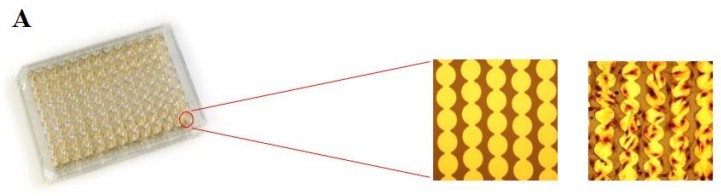

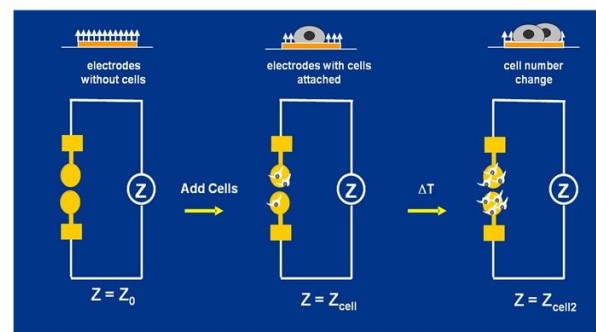

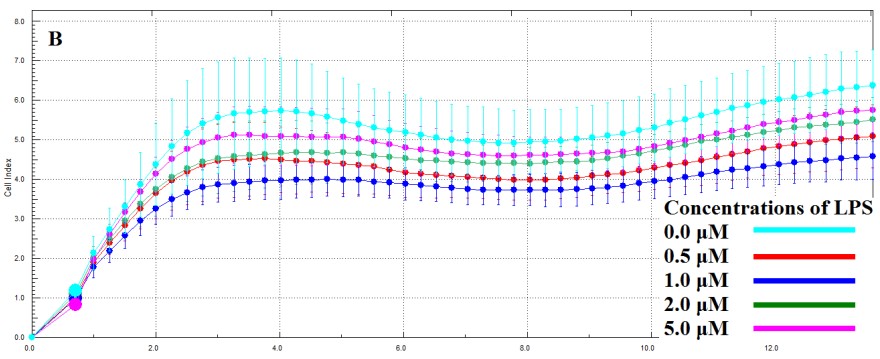

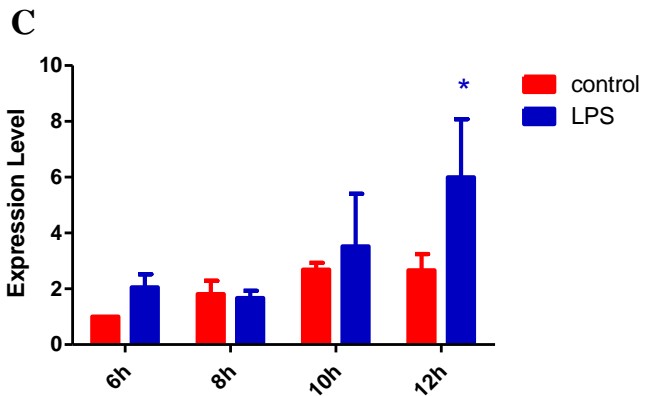

**Figure 1 Real Time Cellular Analysis (RTCA) of micro-electrical impedance changes of HPAECs.**
(A) A schematic showing the experimental design. (B) Micro-electrical impedance of HPAECs detected in real-time after 12 h of culture with different final concentrations of LPS. (C) Histogram showing the change in TNF-$\alpha$ expression in HPAECs, at different time points using RT-PCR, in the presence of 1 $\mu$M LPS (*$p < 0.05$).

## The effect of MSCs on acute injury of HPAECs

HPAECs were incubated in a 24-well plate with 1 μM of LPS and were divided into four groups, based on the treatment timeline, control group, 8 h, 16 h and 24 h treatment groups. Following this the cells were non-contact co-cultured with MSCs. The cells were collected and the change in expression of TNF$\alpha$ was examined by RT-PCR. Results show that the effective treatment window of acute endothelial cell injury by MSCs is 8h (Fig. 2A).

In addition, the treatment groups were divided into another four groups based on the cell ratio as, control group (HPAECs only), groups with inoculation ratio of HPAECs and MSCs 1:1, 1:2 and 1:4. They were non-contact co-cultured with HPAECs following treatment with LPS. Cells of each group were collected and the change in expression of TNF$\alpha$ was examined by RT-PCR. Results show that larger ratio of MSCs is not necessarily better. Best treatment effect was achieved when the ratio of HPAECs and MSCs was 1:2 (Fig. 2B). Results from label-free Real Time Cellular Analysis (RTCA) also show that, compared to that of the control group, the micro-electrical impedance of groups where HPAECs were co-cultured with MSCs were significantly higher and reached their peak after 10 h. Among them, micro-electrical impedance of the group with the ratio of HPAECs to MSCs 1:2 was the highest. At 15 h later, micro-electrical impedance of the control group was significantly higher than that of the other groups (Fig. 2C). The results were consistent with the change in expression of TNF-$\alpha$, indicating that the effect of MSCs on acute injury of HPAECs was best at a ratio of 1:2 and the best treatment time is about 8 h to 10 h.

## The effect of S1P acute HPAEC injury

Sphingosine-1-phosphate (S1P) is a biologically active ubiquitous sphingomyelin. It is mainly produced by platelets and also present in plasma and tissues. When its physiological concentration is in the range of 0.2 to 1.1 μM, it can effectively enhance the endothelial barrier function. In this study, a HPAEC acute injury model was established using 1 μM LPS and treated with S1P at final concentrations of 0 μM, 0.5 μM, 1 μM, 2 μM, and 5 μM. The effect of LPS on HPAEC injury was studied by RTCA experiment. Results show that, S1P within physiological concentrations can effectively enhance the micro-electrical impedance of HPAECs (Fig. 3A). In addition HPAECs were cultured in a 16-well E-Plate and 1 μM LPS was added to induce injury. Meanwhile, MSCs at the twice concentration was inoculated into a 16-well E-Plate Insert. After incubated for 12 h, the LPS medium was discarded and S1P at concentrations of 0 μM, 0.5 μM, 1 μM, 2 μM, and 5 μM were added in the E-Plate and E-Plate Insert for co-culture, then the effect on HPAEC injury was studied by RTCA. Results show that S1P within physiological concentration (0.5 μM and 1 μM) can effectively enhance the micro-electrical impedance of HPAECs and alleviate LPS induced injury to HPAECs (Fig. 3B). Both sets of data show that the enhancing effect of 0.5 μM S1P on micro-electrical impedance was superior to that of 1 μM S1P in the presence or absence of MSCs. In addition, using a Transwell plate, HPAECs exposed to 1 μM LPS for 12 h were co-cultured with MSCs at a ratio of 1:2 and 0.5 μM S1P for 8 h. HPAECs were collected and the change in expression of TNF-$\alpha$ was determined. Results show that compared to MSCs alone, the down-regulation of TNF-$\alpha$ expression was more significant when S1P was combined with MSCs (Fig. 3C).
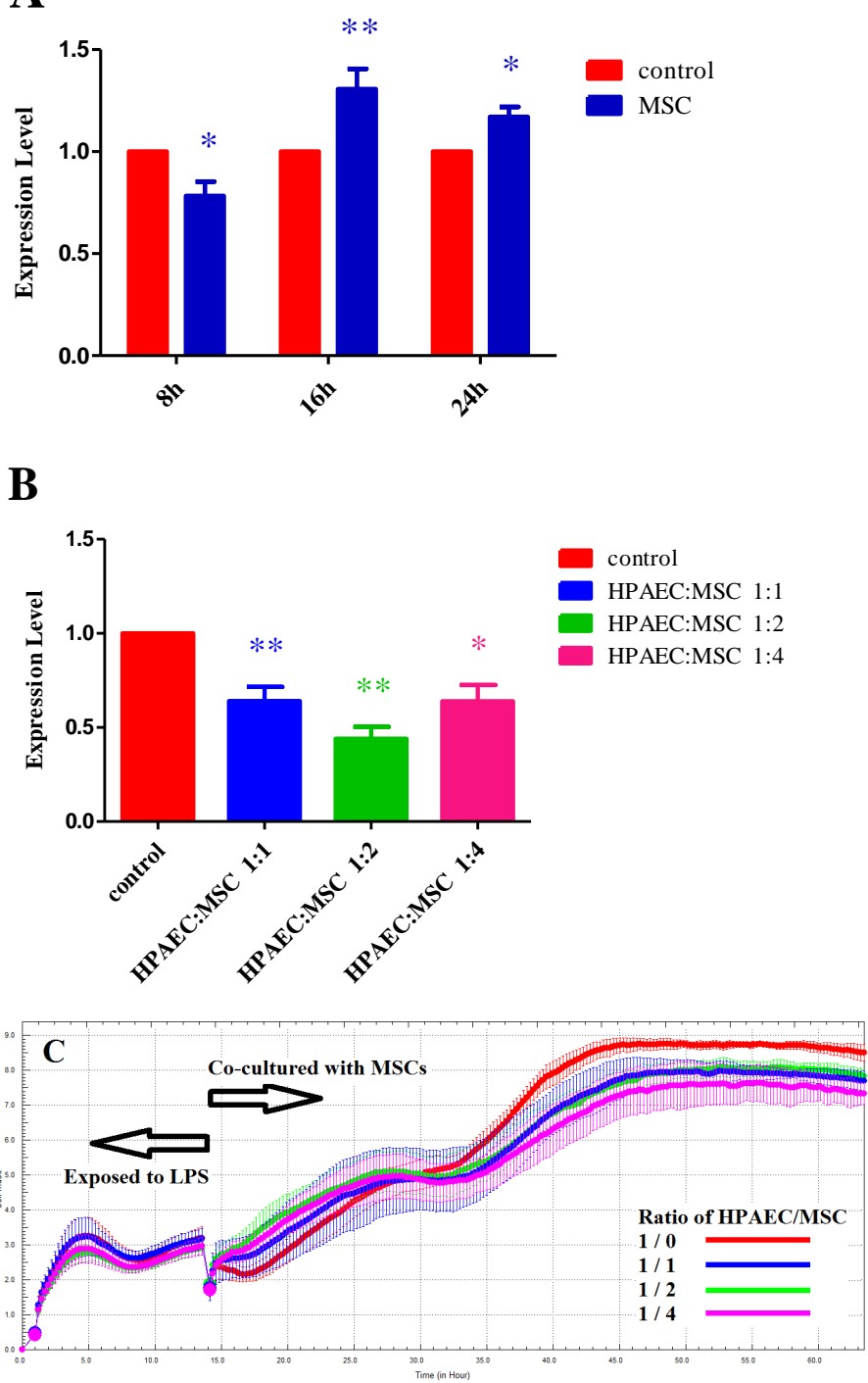

**Figure 2** (A) The change in expression of TNFα at different time points when injured HPAECs were co-cultured with MSCs. Histogram showing TNFα expression in HPAECs, using RT-PCR. (*$p < 0.05$, **$p < 0.01$). (B) The change in expression of TNFα when injured HPAECs were co-cultured with different proportions of MSCs as examined at different time points by RT-PCR (*$p < 0.05$ and **$p < 0.01$). (C) The micro-electrical impedance of HPAECs exposed to LPS and co-cultured with MSCs as examined using RTCA system.

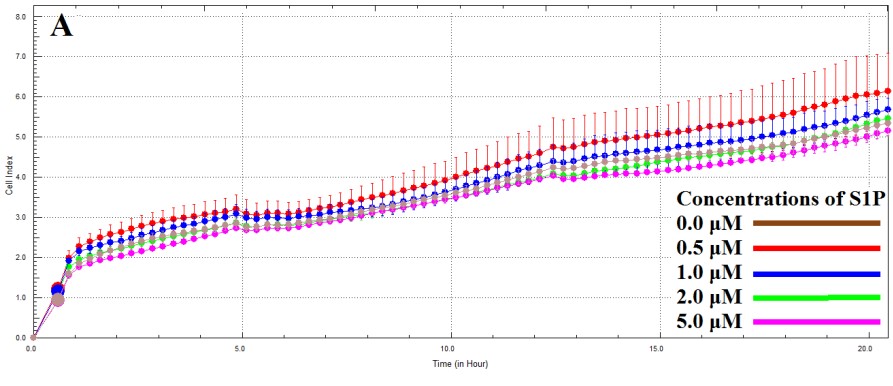

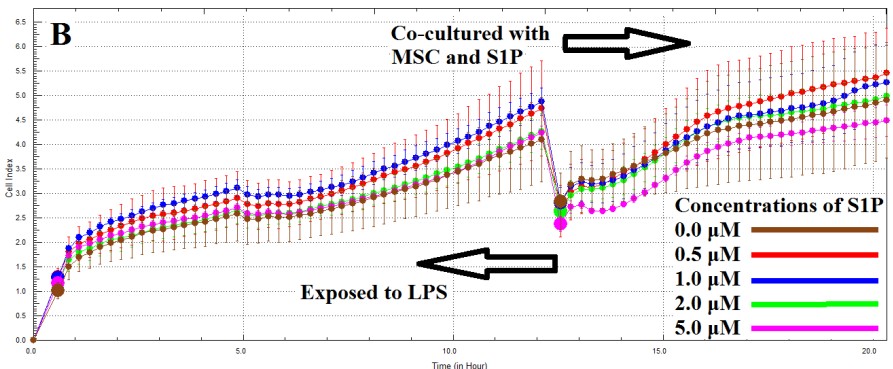

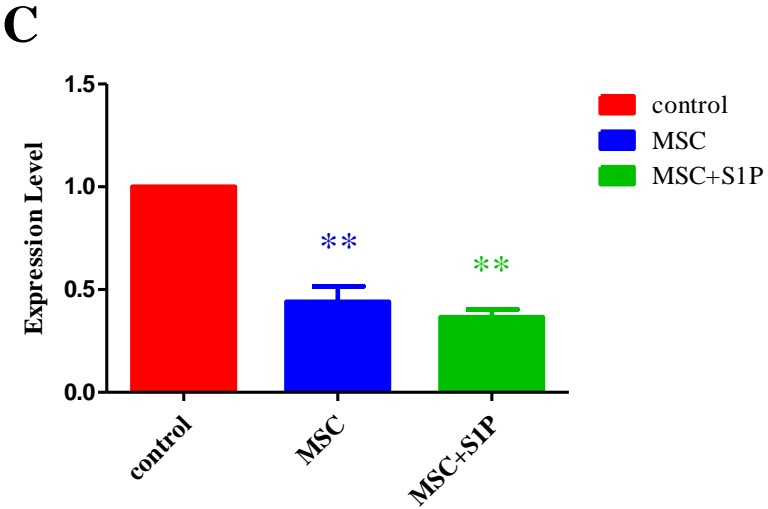

**Figure 3  The combined effect of LPS and S1P on micro-electrical impedance of HPAECs.** (A) The effect of S1P on HPAEC injury studied using RTCA. Results show that, S1P within physiological concentration (0.5 μM and 1 μM) can effectively enhance the micro-electrical impedance of HPAECs. (B) The combined effect of MSCs and S1P on micro-electrical impedance of LPS injuried HPAECs as examined using the RTCA system. (C) Change in expression of TNFα in injured HPAECs when treated with S1P and MSCs using RT-PCR (**$p < 0.01$).

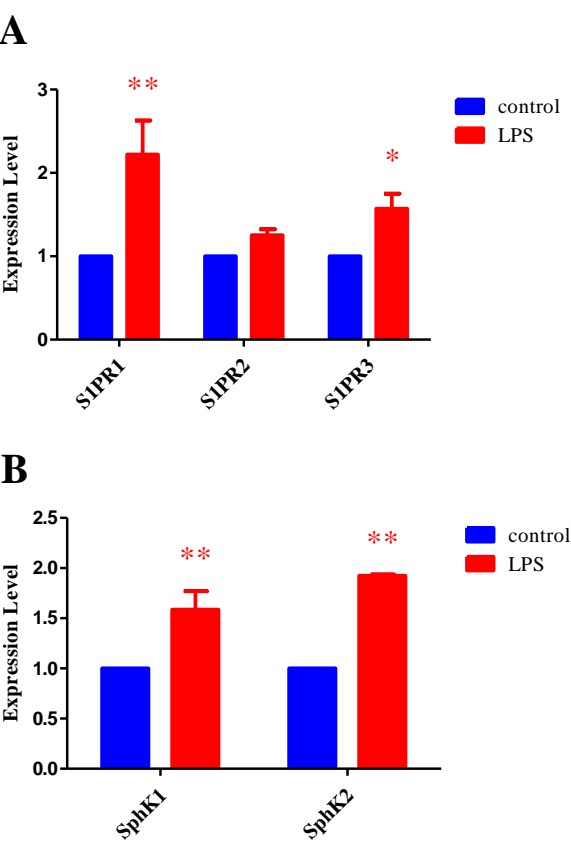

**Figure 4** **S1P receptor and Sphingosine kinase expression following acute injury.** (A) Representative figure showing expression change of S1P receptors 1, 2 and 3. (B) Histogram demonstrating the expression change of sphingosine kinases 1 and 2 ($*p < 0.05$, $**p < 0.01$).

## Effects of S1P and MSCs combination therapy compared to MSCs alone

In order to further investigate the possible synergistic mechanism of MSC and S1P in the treatment of HPAEC injury, the change in S1P receptors and sphingosine kinase expression in HPAECs treated with MSCs alone or the combination of MSCs with S1P, were determined using RT-PCR. After 12 h following exposure to LPS, the expression changes of S1P receptors (S1PR1, S1PR2 and S1PR3) and sphingosine kinases (SphK1 and SphK2) in HPAECs were studied using RT-PCR. Results demonstrate that the expression of target genes were dramatically increased following LPS induced injury. The expression of S1PR1 and S1PR3 were increased significantly, while that of S1P2 was not. There was no significant difference in the expression levels of SphK1 and SphK2 following LPS induced injury (Fig. 4). These results indicate that the S1P related genes could act as potential therapeutic targets. It also suggests that acute lung injury is a complex pathophysiological process and the treatment directed to a single target may not achieve the best efficacy. The combination of MSCs and S1P may become an effective therapy.

Furthermore, after 12 h of LPS exposure, the medium was changed. According to the co-culture condition, HPAECs were divided into 3 groups, that is, control group, MSC

therapy group and combined therapy group (with 0.5 μM S1P and MSCs at a ratio of 1:2 of HPAECs). After 8 h of non-contact co-culture, HPAECs cells were collected and total RNA were extracted. The changes in expression of S1P related genes were detected using RT-PCR. Results show that compared with MSC alone, the down-regulation of TNF-$\alpha$ expression was significant when MSCs were used in combination with S1P. This indicated that there is a possible synergistic mechanism, between MSCs and S1P, in the treatment of acute injury (Fig. 5A). The regulation results of S1P receptors show that when MSCs work on injured HPAECs, its regulation of S1P receptors were different, affecting only S1PR2 and S1PR3. The expression of S1PR2 and S1PR3 were further reduced when MSCs were used in combination with S1P. There was no significant difference regulatory effect on S1PR1 irrespective of whether MSCs were used alone or in combination with S1P (Fig. 5B).

Studies have shown that the intracellular level of S1P is tightly regulated by the balance between synthesis and degradation. *In vivo*, sphingomyelin (SM) is catalyzed to produce ceramide, which is further hydrolyzed to sphingosine. Sphingosine is able to produce S1P by phosphorylation catalyzed by sphingosine kinase (SphK1 and SphK2). Our study results demonstrate that the expression of sphingosine kinase (SphK1 and SphK2) differs in HPAECs following LPS injury. When MSCs were used alone, it mainly worked on SphK1, the expression level of which was significantly reduced. When MSCs were used in combination with S1P, the expression of SphK2 was also significantly reduced (Fig. 5C). Through analyzing the expressions of S1P related genes, we found that when MSCs were combined with S1P, their effect was not simply the superimposition of each other, but more significant. This suggests that there might be some synergistic mechanism between MSCs and S1P in the treatment of acute injury.

### Impact of MSC and S1P on S1P receptor expression in acute injury

After 12 h of HPAECs exposed to LPS were divided into three groups, based on co-culture conditions as, control group, MSC therapy group and combined therapy group (0.5 μM S1P and MSCs at a ratio of 1:2 to HPAECs). After 8 h of non-contact co-culture, HPAECs cells were collected and total proteins were extracted. The changes in expression of S1P receptors were detected through Western-Blot. The results were consistent with that from the RT-PCR. The results of regulation of different S1P receptors were different when MSCs were added, where S1PR2 and S1PR3 were the main targets. The effect on S1PR1 was not significant whether MSCs were used alone or in combination with S1P (Fig. 6).

## DISCUSSION

Sphingosine-1-phosphate (S1P) is an agonist mainly present in plasma and tissues. It is an important regulator of vascular endothelial cell permeability and fluid balance *in vivo* and is able to enhance endothelial barrier. Its homeostasis *in vivo* is tightly regulated by the balance between synthesis and degradation. *In vivo*, sphingomyelin (SM) is catalyzed by Sphingomyelinase to produce ceramide (Cer), which produces sphingosine in hydrolysis catalyzed by Neuraminidase. Phosphorylation of sphingosine produces S1P catalyzed by sphingosine kinase (SphK1 and SphK2). Physiological concentrations of S1P plays an important role in maintaining endothelial barrier function. It has been demonstrated

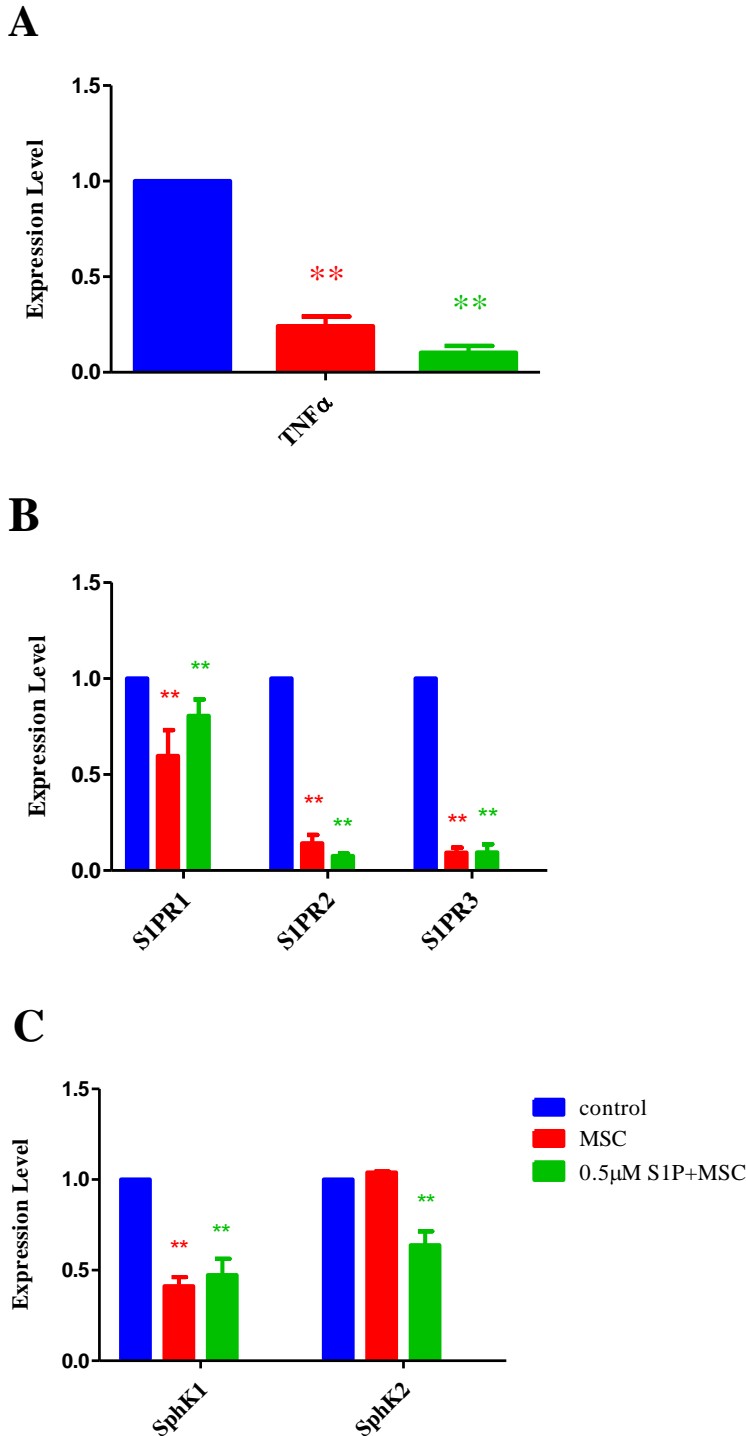

**Figure 5** **Effect of MSC and S1P combination therapy on TNF-α, S1P receptors and Sphingosine kinases.** (A) Histogram demonstrating the expression change of TNF-α. (B) Histogram demonstrating the expression change of S1P receptors 1, 2 and 3. (C) Histogram demonstrating the expression change of sphingosine kinases 1 and 2 ($**p < 0.01$).

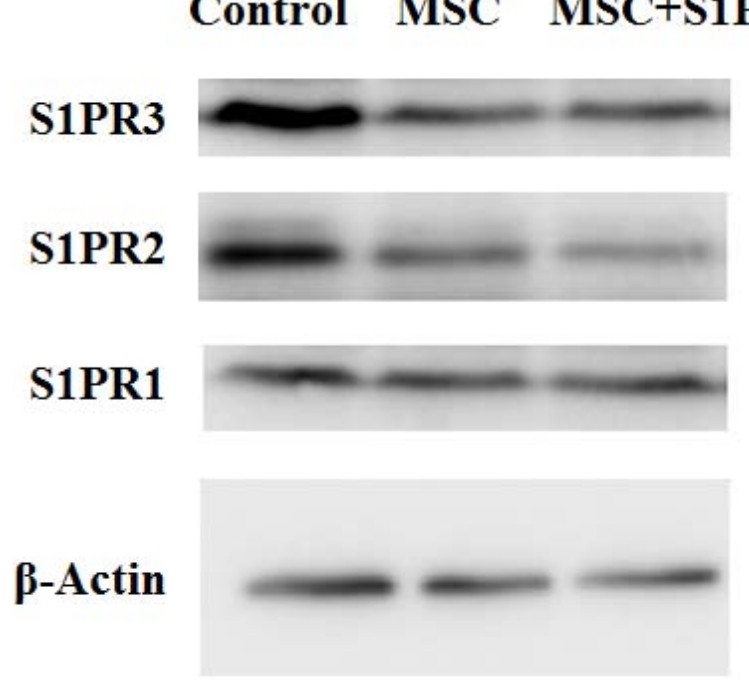

**Figure 6** **S1P receptors expression in HPAECs treated with MSCs.** Pictograph showing Western blot analysis of S1P receptor expression HPAECs exposed to LPS that were treated with MSCs alone or in combination with S1P.

that the activity of SphK1 and the dynamic concentration balance of SphK1/S1P axis, are necessary for the modulation of inflammatory signals and are related to the regulation of innate, adaptive and other immune cells. Studies have also shown that abnormal changes in SphK1 or S1P lead to many inflammatory and autoimmune diseases, including asthma, rheumatoid arthritis, sepsis, inflammatory bowel disease and so on (*Spiegel & Milstien, 2011*). Our study found that in LPS induced acute injury of HPAECs, the expression of SphKs was increased dramatically. When MSCs were used alone, only the expression of SphK1 was reduced, and there was no significant impact on the expression of SphK2. When MSCs were used in combination with S1P, the expression of SphK1 and SphK2 were reduced simultaneously. This shows that MSCs play a role in the treatment of acute lung injury by regulating the expression and activity of SphKs. It also suggests that a possible synergistic mechanism may exist, between MSCs and S1P, in the treatment of acute HPAEC injury.

Studies on S1P show that endothelial barrier enhancement mediated by S1P is completed by activating the Gi and Racl signaling pathway through the S1P receptor (*Yu, Nishi & Kawahara, 2012*). There are five different types of S1P receptors (S1PR1-5) and their distribution and function are different. Those in endothelial cells are mainly S1PR1-3. Wherein the S1P receptor 1 (S1PR1) has a more important biological role in enhancing endothelial barrier, the S1PR2 suppresses the endothelial barrier function (*Blaho & Hla, 2014*). Studies have shown that when the selective S1PR1 competitive antagonist is used in experimental mice, pulmonary endothelial cells integrity is compromised (*Rosen et al., 2007*; *Sanna et al., 2006*). Besides, it has also been found that stimulation of nitrification and

release of S1PR3 in HPAECs helps to suppress the endothelial barrier function (*Sun et al., 2012*). These demonstrates that S1P receptors can be the therapeutic targets of ALI/ARDS to improve endothelial cell barrier. In this study We found that when MSCs works on acute injured HPAECs, the regulation of individual S1P receptors were different, which was mainly acted on S1PR2 and S1PR3. The expression of S1PR2 and S1PR3 were further reduced when MSCs were used in combination with S1P. The regulatory effect on S1PR1 was not significant irrespective of whether MSCs were used alone or in combination with S1P. This indicates that MSCs may play a role in enhancing endothelial barrier function by reducing the expression of S1PR2 and S1PR3 and enhancing specificity of S1P receptors.

Clinical studies have shown that any single target and drug are not likely to reverse the severe pathological injury caused by ALI/ARDS and achieve great therapeutic effect quickly. Innovative multi-target synergistic treatment mechanism may be an effective method to solve this problem. In essence, stem cell therapy does not work on a single target or as a single drug in therapy, but on multiple targets based on biological distress signals from the diseased tissues and as multiple therapies. Our study found that when MSCs worked on HPEACs with acute injury, the expressions of S1P receptors and sphingosine kinase were regulated. This indicates that MSCs are able to affect the expression of multiple S1P related genes, improve endothelial barrier and cure acute lung injury. Meanwhile, as an endogenous bioactive molecule, S1P is an important regulator of vascular endothelial cell permeability and fluid balance. In the treatment of acute lung injury, it plays an important role in improving endothelial barrier by regulating the expression of key enzymes in anabolic processes and specific receptors on signaling pathways. Our study results demonstrate that both MSCs and S1P can alleviate acute lung injury. But the combined use of MSCs and S1P shows significant efficacy in the regulation of S1P related gene expression, suggesting that some synergism between MSCs and S1P may exist in the treatment of acute injury. Further studies are need to determine whether MSCs when used in combination with S1P, further improve the endothelial barrier and if a possible synergistic mechanism exists and maybe to provide a novel therapeutic strategy in the treatment of acute lung injury.

## CONCLUSIONS

In this study we built a pulmonary endothelial cell model of acute injury by LPS, and investigated the regulation of S1P receptors and sphingosine kinases expression by MSCs combined with S1P. The study results demonstrated that both MSCs and S1P can alleviate acute lung injury. When MSCs were used alone, only the expression of SphK1 was reduced, and there was no significant impact on the expression of SphK2. When MSCs were used in combination with S1P, the expression of SphK1 and SphK2 were reduced simultaneously. We also found that when MSCs works on acute injured HPAECs, the regulation of individual S1P receptors were different, which was mainly acted on S1PR2 and S1PR3. The expression of S1PR2 and S1PR3 were further reduced when MSCs were used in combination with S1P. These results show that MSCs play a role in the treatment of acute lung injury by regulating the expression and activity of S1P related genes, when MSC combined with S1P there is a possible synergistic mechanism exists.

### Funding

This study was supported by the Beijing Natural Science Foundation (7164284), Innovation Science Research Foundation of 307th Hospital of PLA (FC-2014-01), Public Health Major Project (AWS15J007). The funders had no role in study design, data collection and analysis, decision to publish, or preparation of the manuscript.

### Grant Disclosures

The following grant information was disclosed by the authors:
Beijing Natural Science Foundation: 7164284.
Innovation Science Research Foundation: FC-2014-01.
Public Health Major Project: AWS15J007.

### Competing Interests

The authors declare there are no competing interests.

### Author Contributions

- Huiying Liu conceived and designed the experiments, performed the experiments, analyzed the data, contributed reagents/materials/analysis tools, wrote the paper, prepared figures and/or tables.
- Zili Zhang performed the experiments, analyzed the data, prepared figures and/or tables.
- Puyuan Li, Xin Yuan and Jing Zheng analyzed the data.
- Jinwen Liu performed the experiments.
- Changqing Bai and Wenkai Niu contributed reagents/materials/analysis tools, reviewed drafts of the paper.

### Ethics

The following information was supplied relating to ethical approvals (i.e., approving body and any reference numbers):

The Affiliated Hospital of Military Medical Science Scientific Research Ethics Committee (ky-2015-3-17).

### Data Availability

The raw data has been supplied as a Supplemental File.

### Supplemental Information

Supplemental information for this article can be found online at http://dx.doi.org/10.7717/peerj.2712#supplemental-information.

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
