# Peer review of "Regulation of S1P receptors and sphingosine kinases expression in acute pulmonary endothelial cell injury"

_PeerJ, doi:10.7717/peerj.2712_

## Round 0.1 · original submission · Major Revisions

Dear Dr Liu

Thanks for your submission to PeerJ. Our decision is "Major Revisions". Please address the reviewer comments in your rebuttal and revision

Best Regards
Bingjin Li

Reviewer 1 ·

Basic reporting

No comments.

Experimental design

No comments.

Validity of the findings

No comments.

Additional comments

Liu et al. reported a combination of mesenchymal stem cells (MSC) with sphingosine-1- phosphate (S1P) in suppressing acute injury of HPAECs in vitro, proposing a potential therapy for acute lung injury. While the observation is interesting, there are a few of issues which need to be clarified as follows;
1. The characterization of MSCs used in this study has to be shown.
2. FTY720, as the structure analogue of S1P, is approved by FDA in the treatment of multiple sclerosis. Why not test the effect of FTY720 for the combination treatment?
3. The MSCs medium contains many cellular regulatory factors. In the co-cultured system for HPAECs and MSCs, how to exclude the effect of those factors in MSC culture medium?
4. Some results in the figures have no statistical analysis, particularly in RTCA data.
5. The expression changes of SphKs need to be confirmed by western blot or other methods.
6. Since all experiments have been done in vitro, the conclusion regarding the therapy should be tuned down.
7. The methods for determining endothelial cell injury and barrier function need to be detailed.

Reviewer 2 ·

Basic reporting

The authors reported the regulation of S1PRs and SphKs expression by MSCs used in combination with S1P during the treatment of ALI, and found that when MSCs were used in combination with S1P, the selectivity of S1P receptors was increased and the homeostatic control of S1P concentration was elevated through regulating the expression level of S1P metabolic enzymes. The authors claimed that they established the molecular basis for clinical treatment of ALI using MSCs in combination with S1P and their possible synergistic mechanism.

Experimental design

The research is well designed

Validity of the findings

1. Real Time Cellular Analysis was used to investigate the HPAECs Micro-electronics impedance, however, the results in the figure 1,2 and 3 have no statistical analysis, please show the details of the statistical analysis.
2. The wording and style of some section, particularly in figures, need carefully editing. For example in fig5, the X axis title should be TNF-α, not TNFa.
3. In Fig 6 , the western blot result of β-actin have four lanes, is different with the result of other proteins, what is the reason?
4. The author claimed that when MSCs were used in combination with S1P, the homeostatic control of S1P concentration was improved through regulation of expression of S1P metabolic enzymes, why not investigated the S1P concentration ?

Additional comments

no

---

## Round 0.2 · accepted · Accept

Dear Liu

I am glad to inform you that your manuscript has been accepted.

Best Regards
Bingjin Li

Reviewer 1 ·

Basic reporting

None comments.

Experimental design

No comments.

Validity of the findings

No comments.

Additional comments

The concerns have been addressed. No further comments.

Reviewer 2 ·

Basic reporting

No Comments

Experimental design

No Comments

Validity of the findings

No Comments

Additional comments

No Comments